

**A Machine Learning Approach to Aerosol Classification for Single**
**Particle Mass Spectrometry**
**Christopoulos, Costa D.[1], Garimella, Sarvesh[1,2], Zawadowicz, Maria A.[1,3], Möhler,**
**Ottmar[4] and Cziczo, Daniel J.[1,5]**
[1] Department of Earth, Atmospheric and Planetary Sciences, Massachusetts Institute of
Technology, Cambridge, MA, United States
[2] ACME AtronOmatic, LLC, Portland, OR, United States
[3] Atmospheric Sciences and Global Change Division, Pacific Northwest National
Laboratory, Richland, WA, United States
[4] Institute of Meteorology and Climate Research, Karlsruhe Institute of Technology,
Karlsruhe, Germany
[5] Department of Civil and Environmental Engineering, Massachusetts Institute of
Technology, Cambridge, MA, United States



**Abstract**
Compositional analysis of atmospheric and laboratory aerosols is often conducted via
single-particle mass spectrometry (SPMS), an *in situ* and real-time analytical technique
that produces mass spectra on a single particle basis. In this study, machine learning
classifiers are created using a dataset of SPMS spectra to automatically differentiate
particles on the basis of chemistry and size. Machine learning algorithms build a
predictive model from a training set for which the aerosol type associated with each mass
spectrum is known *a priori*. Classification models were also created to differentiate
aerosol within four broad categories: fertile soils, mineral/metallic particles, biological,
and all other aerosols. Differentiation was accomplished using ~40 positive and negative
spectral features. For the broad categorization, machine learning resulted in a
classification accuracy of ~93%. Classification of aerosols by specific type resulted in a
classification accuracy of ~87%. The 'trained' model was then applied to a 'blind'
mixture of aerosols which was known to to be a subset of the training set. Model
agreement was found on the presence of secondary organic aerosol, coated and uncoated
mineral dust and fertile soil.
**1. Introduction**

19        The interaction of atmospheric aerosols with clouds and radiation contributes to the

uncertainty in determinations of both anthropogenic and natural climate forcing [Boucher
et al., 2013; Lohmann and Feichter, 2005]. Aerosols directly affect atmospheric radiation



by scattering and absorption of radiation from both solar and terrestrial sources. The
radiative forcing from particulates in the atmosphere depends on optical properties that
vary significantly among different aerosol types [Lesins et al., 2002].  Aerosols also
indirectly affect climate via their role in the development and maintenance of clouds
[Vogelmann et al., 2012; Lubin et al., 2006]. Ultimately, the formation, appearance, and
lifetime of clouds are sensitive to aerosol properties like shape, chemistry, and
morphology [Lohmann and Feichter, 2008]. Characterization of aerosol properties,
therefore, plays a vital role in understanding weather and climate.

9         The chemical composition and size of aerosols has been analyzed on a single

particle basis *in situ* and in real-time using single particle mass spectrometry (SPMS;
Murphy [2007]). First developed ~2 decades ago, SPMS permits the analysis of aerosol
particles in the ~150 – 3000 nm size range, while differentiating internal and external
aerosol mixtures and characterizing both volatile (e.g. organics and sulfates) and
refractory (e.g. crystalline salts, elemental carbon and mineral dusts) particle components.
Particles are typically desorbed and ionized with a UV laser and resultant ions are
detected using time-of-flight mass spectrometry [Murphy, 2007]. A complete mass
spectrum of chemical components is normally produced from each analyzed aerosol
particle [Coe et al., 2006]. Despite almost universal detection of components found in
atmospheric aerosols, SPMS is not normally considered quantitative without specific
laboratory calibration [Cziczo et al., 2001].

21        Aerosols with different properties can appear similar in the context of SPMS. For

example, fly ash and mineral dust contain peaks corresponding to silicates, phosphates,
metals, and metal oxides despite different origins and emission sources [Zawadowicz et



al., 2017]. This complicates analysis of aerosol populations because their properties need
to be well-defined in order to increase agreement between models and observations
[Niemand et al., 2012; Hoose and Möhler, 2012; Welti et al., 2009]. Even minor
compositional changes can be atmospherically important. As one example, mineral dusts
are known to be effective at nucleating ice clouds [Cziczo et al., 2013]. Particles in the
atmosphere undergo chemical and morphological changes as they age and eventually
contain material from several sources [Boucher et at. 2013]. Despite minor addition of
mass, aged mineral dust is less suitable for ice formation [Cziczo et al., 2013], but these
particles then act as cloud condensation nuclei and participate in warm cloud formation
[Andreae et al., 2008]. As a second example, ice nucleation in mixed-phase clouds has
been suggested to be predominantly influenced by feldspar, a single component among
the diverse mineralogy of atmospheric dust [Atkinson et al., 2013].
Here we show that supervised training and a rule-based probabilistic classification
of a decision tree ensemble can be used for differentiation of SPMS spectra. Various
clustering methods have been used to group aerosol types [Murphy et al., 2003; Gross et
al., 2008] but these algorithms are known to struggle with chemically-similar aerosols as
they do not incorporate known particle labels in the training process. Such 'unsupervised'
clustering algorithms automatically group unlabeled data points on the basis of a
specified distance metric in feature space, in this case mass spectral signals. For the
purposes of setting broad aerosol categories, which are chemically similar and easily
separable in feature space, clustering is the simpler tool and the data easier to interpret.
For identifying new or potentially unexpected atmospheric aerosols, such properties are
desirable; however, the advantages of clustering greatly diminish when considering



similar particle types that overlap in feature space. Fertile soils, for instance, are often
grouped into a single category despite different sources and atmospheric histories.
Clustering algorithms should therefore be considered as a tool to use alongside
supervised classification. The latter may be used to further explore unique aerosol types
or verify manually labeled clusters with higher precision. Furthermore, the ensemble
approach presented here also produces variable rankings and probabilistic predictions that
assist in addressing measurement uncertainty.

8         In this study, we demonstrate the capabilities of machine learning to automatically

differentiate particles on the basis of chemistry and size. The resulting model can capture
minor compositional differences between aerosol mass spectra. By testing predictions
using an independent, or 'blind', dataset, we illustrate the feasibility of combining on-line
analysis techniques such as SPMS with machine learning to infer the behavior and origin
of aerosols in the laboratory and atmosphere.
**2. Methodologies**
**2.1 PALMS**

16        The Particle Analysis by Laser Mass Spectrometry (PALMS) instrument was

employed for these studies. PALMS has been described in detail previously [Cziczo et al.
2006]. Briefly, the instrument samples aerosol particles in the size range from ~200 to
~3000 nm using an aerodynamic lens inlet into a differentially-pumped vacuum region.
Particle aerodynamic size is acquired by measuring particle transit time between two 532
nm continuous wave neodymium-doped yttrium aluminum garnet (Nd:YAG) laser beams.
A pulsed UV 193 nm excimer laser is used to desorb and ionize the particles and the



resulting ions are extracted using a unipolar time-of-flight mass spectrometer. The
resulting mass spectra correspond to single particles. The UV ionization extracts both
refractory and volatile components and allows analysis of all chemical components
present in atmospheric aerosol particles [Cziczo et al. 2013].
**2.2 Dataset**

7        A set of 'training data' was acquired by sampling atmospherically-relevant

aerosols. The majority of the dataset was acquired at the Karlsruhe Institute of
Technology (KIT) Aerosol Interactions and Dynamics in the Atmosphere (AIDA) facility
during the Fifth Ice Nucleation workshop — Part 1 (FIN01). The remainder were
acquired at our Aerosol and Cloud Laboratory at MIT. The FIN01 workshop was an
intercomparison effort of ~10 SPMS instruments, including PALMS. The training data
correspond to spectra of known particle types that were aerosolized into KIT's main
AIDA and a connected auxiliary chamber for sampling by PALMS and the other SPMSs
(Table 1). Hereafter we group both chambers with the name 'AIDA'. The number of
training spectra acquired varied by particle type, ranging from ~250 for secondary
organic aerosol (SOA) to ~1500 for potassium-rich feldspar ("K-feldspar"). In total,
~50,000 spectra are considered with each spectrum containing 512 possible mass peaks
and an aerodynamic size. (Table 2). Additionally, the FIN01 workshop included a blind
sampling period, where AIDA was filled with 3 - 4 aerosol types known to be from the
training set (i.e., for which spectra had already been acquired) but (*a priori*) of unknown
size, specific types and at unknown concentrations.



Figure 1 illustrates a simple differentiation of particles using only two mass peaks
in one (negative) polarity. Mass peaks represent fractional ion abundance, measured as a
normalized total signal (ion current). In this example, the normalized areas of negative
mass peaks 24 ($C_2^-$) and 16 ($O^-$) are plotted. Distinct aerosol types are differentiated by
color with clusters forming in this two-dimensional space. Note that spectra of the same
aerosol type form distinct clusters (e.g. Arizona Test Dust, ATD), as do similar aerosol
classes (e.g., soil dusts). Co-plotted in Figure 1 are data from the blind experiment.
Distinct clusters of spectra from the blind experiment are noticeable and correlate with
known clusters.    Described in the next section, machine learning algorithms draw
"decision boundaries" that best separate different groups of data points based on set of
rules. Machine learning is not bound by the simplistic two dimensional space shown in
Figure 1 and instead uses all 512 mass peaks and aerodynamic size.
**2.3 Aerosol Classification**
A trained classification model maps a continuous input vector 'X' to a discreet
output value using a set of parameters 'learned' from the data. Figure 2 illustrates the
mapping of a mass spectrum to vector space. In contrast to traditional, hard-coded, rule-
based classification methods, machine learning determines parameters that partition the
data set. To form X, mass spectra are converted to dimensional vectors normalized to the
total ion current (i.e., the total of all mass peaks sum to 1 in each spectrum). The elements
of the vectorized mass spectrum, termed 'features', hold information about the ionization
efficiency and relative abundance of chemical species in each aerosol and serve as the
variables for the machine learning model.



1       Machine learning is conducted in two phases: training and testing. During training,

a model is constructed and iteratively updated based on data (i.e., mass spectra) from the
training set. For this work, the set of known aerosol types sampled by PALMS was
converted to dimensional vectors. These data form the basis set for defining each aerosol
type. An ensemble of decision trees was used to generate predictions of aerosol type. A
single decision tree is a statistical decision model that performs classification based on a
series of comparisons relating a variable $X_i$ (in this case a normalized mass peak in X) to
a learned threshold value [Breiman, 2001]. Represented as an algorithmic tree, a binary
decision tree consists of a hierarchy of nodes where each node connects via branches to
two other nodes deeper in the tree. At each node, one of the two branches is taken based
on whether a normalized peak $X_i$ is greater or less than a threshold value. Each branch
leads to another node where a different test is performed. After a series of tests, one at
each node, a class is assigned to a given sample; these are the so-called 'leaves'. Figure 2
illustrates the classification model for a single decision tree.

15       Each test in the tree narrows the set of reachable output leaves and thus the

sample space of possible aerosol labels. After $h$ tests in this study, where $h$ ranges from
10 to 3000, the set of reachable leaves and possible labels is 1 and the decision tree
outputs a prediction. Because PALMS is unipolar – either a positive or negative mass
spectrum is produced – simultaneous generation of positive and negative spectra on a
particle-by-particle basis is not possible.  Two separate classification models, one for
each polarity, were therefore generated to classify aerosols. These are hereafter referred
to as the 'positive' and 'negative classification algorithms'.





**2.4 Decision Tree Ensembles**
An ensemble consists of a collection of classifiers where each independently
labels a spectrum vector X. To make a final prediction of aerosol type, decision trees
within an ensemble 'vote' on a classification label. Each vote has equal weight and the
spectrum is assigned to the majority choice. Each tree within an ensemble is
independently grown on a subset of the training data so that a commonly voted label
implies a higher certainty. Adding members to an ensemble increases the robustness of a
classification model by providing alternative hypotheses and is therefore preferable to
single classifiers.
Before an ensemble method is implemented for classification, trees are
independently grown during training.  A total of $k$ trees, with $k = 1000$, were grown using
a bootstrap sample from the training set. In bootstrap sampling, each tree sees an
independent sample set of equal size drawn from the full training set by sampling spectra
with replacement. On average, each tree is built with ~63% of the data. The unsampled
data, known as 'out-of-bag' observations, provide a means to assess classification error
for each tree during the training process.
Given a bootstrap sample, a binary decision tree is grown by sequentially creating
tests that maximize the separation between classes in parameter space.  A test is created
by defining a comparison that minimizes the information entropy of a possible split, thus
minimizing the randomness of prediction labels [Breiman, 1996]. To generate variability
in the model, a best split is chosen among a random set of possible splits at each node on
the basis of entropy [Breiman, 2001]. After iteratively defining thresholds for each new





node, the tree grows in size until a series of tests ending at some node $S_q$ uniquely
characterizes an aerosol as a particle type. A leaf is then appended to node $S_q$ with the
corresponding label. In classification mode, an aerosol spectrum that passes the same tree
will undergo the same series of tests and will end in the same leaf, thus being labeled in
the same way. For the purposes of this study, each tree had ~3,300 nodes.
**2.5 Dimensionality Reduction and Chemical Feature Selection**
Dimensionality reduction is the process of representing data with fewer variables
than initially present in the dataset, in this case less than the original 512 mass peaks and
aerodynamic size. In addition to facilitating data visualization, reducing computation time
and limiting overfitting [Mjolsnes, 2001], dimensionality reduction, in the context of
aerosol mass spectra, also indicates the most important chemical makers for
differentiation. Feature ranking was algorithmically determined by comparing the
performance of trees before and after removing information about peak $X_i$. The method is
that the values of variable $X_i$ is permuted for tree $k$ in the out-of-bag set so that the
variable is irrelevant to the final label. The change in misclassification before and after
the permutation is calculated and then repeated for all trees so that a variable ranking is
obtained [Breimann, 2001]. Table 2 rows ranks mass peaks (features) by polarity in
importance using this method. The columns at left list feature rankings (i.e., most to least
important for correct classification) for the entire set of aerosol types. The columns at
right list rankings when aerosol types are grouped into the broad, chemically similar,
categories. A final ranking was determined by sequentially adding variables and
observing classification performance response. All variables preceding two e-foldings in
classification error were maintained in the final model. Both the specific aerosol type and





broad aerosol category models were retrained using this subset of the initial variables,
listed in Table 2.

3        It is noteworthy that while most of the features are logical differentiators of the

aerosol types investigated in FIN01 there were also surprises. One example is $59^{+}$
(cobalt), determined to be one of the most important features for differentiation. Further
investigation determined this material was a contaminant from dry powder dispersion
equipment used on some samples. This serves to illustrate the lack of *a priori* judgment
by the algorithm and an unintended benefit of machine learning process (i.e.,
contamination identification).

10       **3. Results**

**3.1 Confusion Matrices and Probabilistic Model Performance**

12       A confusion matrix captures misclassification tendencies by pair-wise matching

the model prediction with the true aerosol type or broad category [Powers, 2007].
Confusion matrices represent model predictions as columns $i$ and true aerosol type of
category as rows $j$, where class names are mapped to integers $i, j \in \{1, 2, \dots, y\}$. In this
study, matrices have been normalized along each column to show the fraction of aerosols
labeled as $j$ that actually belong to $i$ (Figures 3 and 4). For aerosol classification, these
matrices can also be interpreted as similarity measures between particle types. Since the
basis of decision tree classification is mathematical separation of physical quantities,
misclassifications result from similarity in mass peaks and their ion abundance between
aerosol types. This is most easily visualized as overlapping clusters in the simple two
dimensional space in Figure 1.





Because the size of the set is large (~22,300), the general classification behavior
can be quantified in term of conditional probability. If $\hat{Y}_i$ is the set of predicted aerosol
spectra with aerosol label $i$ and $Y_j$ is the corresponding set of true spectrum-label pairs for
label $j$, then the conditional probability of assigning an aerosol to label $i$ given a predicted
label $j$ is given by:
$$p(i \mid j) = \frac{|Y_j \cap \hat{Y}_i|}{|Y_j|}$$    (1)
C is the raw confusion matrix of spectrum counts and $p(i \mid j)$ is the conditional
probability distribution over all true aerosol labels $i$, conditioned on some model-
generated label $j$. To obtain matrix P, which encodes $p(i \mid j)$ for all possible labeling,
columns of C are normalized with respect to the total aerosol counts for each label with
Eq. 1.
Model performance for each aerosol is summarized in the diagonal elements of P,
which represent the fraction of aerosol in column j labeled correctly. The classification
accuracy ($a$) is given by averaging diagonal elements of P. A perfect classification model
produces the identity matrix, as all data points are classified correctly 100% of the time.
For example, in the positive confusion matrix, SOA and Agar growth medium are
correctly labeled in the test set 100% of the time. Barring element truncation, all columns
of P add to 1.
Figures 3 and 4 display confusion matrices as heat maps for the full set of particle
labels and broad grouped particle categories, respectively. Broad categories are
delineated by bold horizontal and vertical lines in Figure 3 as fertile soil (Argentinian,
Chinese, Ethiopian, Moroccan and two German soils), pure mineral dust and metallic
particles (ATD, illite NX, fly ash, Na-feldspar, K-feldspar), biological (Agar growth





medium, *P. syringae* bacteria, cellulose, Snomax, and hazelnut pollen), and other (K-
feldspar with sulfuric acid (SA) and SOA coatings, soot, and SOA) particles. Some
model confusion exists between fertile soils and coated/uncoated feldspars which can be
explained since soils are mineral dust mixed with organic and other materials.

5        Positive mass spectra appear to hold more information with respect to

differentiating aerosols than negative. Label-wise classification accuracy for the negative
algorithm ranges from 3-5% lower. A large part of this performance discrepancy is due to
greater ability of positive spectra to differentiate coated particles within the 'other'
category.

10       In addition to quantifying misclassification tendencies between classes, the

confusion matrix can be redefined to show confusion for aerosols within broad categories
themselves. Intraclass misclassification analysis is accomplished by considering smaller
portions of C and using the same probabilistic assumptions highlighted for the full
confusion matrices to form modified probability distributions. The full confusion matrix
is partitioned into submatrices representing confusion in a specific aerosol category and
renormalized with respect to matrix columns. L is the subset of particle labels of a
broader set of aerosols.  Integrating the full conditional probability distribution over
labels that are impossible to observe gives the probability distribution over members of
L:

$$P_l(i,j) = p(i \in L \mid j \in L) = \frac{C(i \in L, j \in L)}{\sum_{i' \in L} C(i', j \in L)} \qquad (2)$$

20       For example, to determine $P_l(i \mid j)$ for fertile soils, a submatrix is formed by

collecting spectral counts in the first 6 rows and columns of the full confusion matrix
(Figure 3). Column normalization is then applied to derive a probability distribution over



labels in the fertile soil category, conditioned on the aerosol actually being a fertile soil.
This analysis is repeated over all categories in both models. Finally, the relative
performance of both models is isolated and considered with respect to each specific
aerosol category.
The precision score [Powers, 2007] captures the classification behavior for some
subset of aerosol L by averaging fractions of correctly classified aerosols for labels
within that category:
$$\text{Precision Score(L)} = \frac{1}{|L|}\sum_{i=j}^{|L|} P(i \in L, j \in L) \qquad (3)$$
When applied to $P_l$, the precision score captures classification performance on a
population with only aerosol labels contained in L. The algorithm is expected to correctly
label an aerosol in such a population with a probability equal to the precision score. The
precision score is valuable when using the classification model as a particle screener,
producing probability distributions over a subset of aerosol labels of interest. The
confusion characteristics are shown in Table 3 for each category in terms of the precision
score and the mean and standard deviation of misclassification within each category.
Although both models perform similarly for biological spectra, discrepancies of 2-5%
appear in the remaining categories. For regimes consisting of only mineral/metallic or
other particles, the positive algorithm shows intraclass performance advantages in terms
of the precision score, but most notably in terms of fewer mislabeling of mineral/metallic
particles.  The largest precision discrepancy is observed for fertile soils, where the
positive ion algorithm has a 5% advantage in precision with approximately half the false
labeling rate.



**3.2 Characterization of Blind Data**
As part of the FIN01 workshop, it was known that 3 - 4 aerosol types from Table
1 were aerosolized into the ADIA chamber but at unknown size and relative
concentration. PALMS, one member of the blind intercomparison effort, collected
~25,000 spectra. After data analysis, the aerosol types and relative abundances were
provided to each group (Figure 5, top center).
The presence or absence of particle types in the blind set was initially diagnosed
by choosing particles predicted at or above the 1% level. We note here that this step was
based on the knowledge that (1) a distinct set of particles would be placed in the chamber
and (2) particles present at or below the 1% level were most likely contamination. We
further note that this step is unique to a blind study and would not be applicable to the
atmosphere. Normalized confusion matrices were redefined for the aerosols in the
population (i.e., those above the 1% level), which forms the labels of set L in Eq. 2.
Finally, particle counts are re-computed by reassigning particle labels based on the
modified confusion matrix. For each particle label $j$, a fraction $n' = P(i \mid j)$ of particles
labeled as $j$ are reassigned to $i$. This probabilistic correction accounts for aerosol
mislabeling tendencies observed during testing, producing statistics that better represent
the underlying aerosol population. The expected fraction of particles belonging to label $i$
(denoted $\hat{n}i$) is given by:
$$< \hat{n}i > = \frac{<n_i>}{|n|} = \frac{1}{|n|}\sum_j P(i \mid j) \, |n_j| \qquad (4)$$
where $n$ is a set containing all blind spectra and $n_j$ is the set of particles labeled as $j$.
Figure 5 illustrates the results after this step, where the bottom charts show
corrected fractional percentages for each aerosol category. Because SOA was nearly


always labeled correctly (Figure 3), the remaining aerosols are considered separately
using the full set of candidate aerosol labels. Both positive and negative models arrived at
similar results, with inconsistencies primarily associated with the presence of trace fertile
soils and mineral dust / fly ash particles. The positive algorithm identifies ~2-4% of the
AIDA population as each Argentinean soil, German soil, ATD, and cellulose whereas the
frequency of these aerosols was too low to consider in the negative. Alternatively, the
negative model estimates Na-Feldspar at ~8% of the total population, a label not
identified by the positive algorithm. This discrepancy can be explained by the 1%
selection criterion for aerosols present in the population. Fertile soils, ATD, and cellulose
frequently accumulate error along rows in the full positive confusion matrix, indicating
frequent confusion with other categories (Figure 3). Furthermore, with the observed
misclassification rates ranging ~1-4%, it is expected that these aerosol labels are false
positives. The negative model offers an alternative hypothesis, suggesting these
miscellaneous aerosols are Na-feldspar. Since there is significant model agreement on the
percentages of SOA, K-Feldspar, and coated feldspars, this part of the blind mixture
population (~90%) can be characterized with most certainty. For the disputed aerosol
labels, more credence is lent to the negative classification algorithm on the basis of
improved precision for fertile soils.

19        The aerosols reported in the blind mixture were soot, mineral dust, and SOA. This

mineral component was not defined and may have been either a specific mineral or soil
dust. The soot aerosols were below the cutoff diameter for PALMS; they were therefore
not detected or identified by the algorithms. Similarly, particles with diameters greater
than ~1000 nm are detected with increasingly large inefficiency which likely leads to



undercounting of mineral dust [Cziczo et al., 2006]. Both algorithms robustly labeled
SOA with large agreement, consistent with the 100% accuracy observed in the test set.
SOA coated mineral dust was identified as a particle type. This material was not
directly input to AIDA but the report is most likely correct, due to coagulation within the
AIDA chamber during the course of the blind experiment. The training data set did not
contain coagulated SOA and mineral dust but did include SOA-coated K-Feldspar, which
explains the identification.
While both models identified a variety of fertile soils, and not a single type, these
results are largely consistent with the known uncertainties highlighted by the confusion
matrices discussed previously. Given the presence of any single mineral dust, some
confusion with fertile soils, SA coated Feldspar, and Na-Feldspar is expected (Figure 3).
Moreover, as discussed previously [Gallavardin et al., 2008], AIDA backgrounds are not
completely particle-free. During the FIN01 study, contamination particles from previous
test aerosol were frequently observed as background and they could also be the origin of
some low-concentration particles matching fertile soil chemistry.
**4. Conclusions and Future Work**
The machine learning approach described here allows for differentiation of
aerosols within a SPMS dataset, augmenting existing tools and reducing the need for a
qualitative comparison between mass spectra. This study lays out a framework for
training and implementing an ensemble classification model and interpreting results in
the context of laboratory and atmospheric aerosol populations. Across a representative
sample of possible aerosol types, the behavior of each algorithm predictably allows users



to infer the presence or absence of specific aerosols and quantify aerosol abundance.
Machine learning is automated and the output of the model must then be informed by
human knowledge of aerosol chemistry. Machine learning should therefore be considered
as an additional tool to interpret mass spectra to better distinguish aerosols with unique
properties in terms of atmospheric chemistry, biogenic cycles, and population health.
The ensemble decision tree classification framework described here may be
generalized to any instrument, or set of instruments, capable of collecting physical and
chemical information that distinguishes particles. Although the method described here is
applied to a stand-alone SPMS and tested with a set of 'blind' data, ancillary laboratory
or field data can be integrated to expand the data set. The success of these algorithms is
data-dependent, where better performance is expected for instruments that provide more,
and more quantitative, analysis of the aerosol properties. Although the algorithms
implemented in this study were primarily used to categorize SOA, mineral dust, fertile
soil and biological aerosols, these models can adopt an arbitrary large set of aerosol data.
**Acknowledgements**
We thank the FIN01 and AIDA teams for logistical support and scientific discussions.
We acknowledge funding from NSF which allowed our participation (grant AGS-
1461347). M.A.Z. acknowledges the support of NASA Earth and Space Science
Fellowship and D.J.C. acknowledges the support of Victor P. Starr Career Development
Chair.



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

| Aerosol type | FIN Label | Description and/or supplier | Generation method | Sample origin | Reference |
|---|---|---|---|---|---|
| Argentinian | SDAr01 | Soil dust collected in La Pampa province, Argentina | Dry-dispersed | KIT | (Steinke et al., 2016) |
| Chinese | SDMo01 | Soil collected from Xilingele steppe, China/Inner Mongolia | Dry-dispersed | KIT | (Steinke et al., 2016) |
| Ethiopian | VSE01 | Soil collected in Lake Shala National Park, Ethiopia (collection coordinates: 7.5 N, 38.7 E) | Dry-dispersed | KIT | N/A |
| German | SDGe01 | Arable soil collected near Karlsruhe, Germany | Dry-dispersed | KIT | (Steinke et al., 2016) |
| Moroccan | DDM01 | Soil collected in a rock desert in Morocco (collection coordinates: 33.2 N, 2.0 W) | Dry-dispersed | KIT | N/A |
| Paulinenaue | N/A | Arable soil collected in Northern Germany (Brandenburg) | Dry-dispersed | KIT | N/A |
| ATD | N/A | Arizona Test Dust, Powder Technology, Inc. (Arden Hills, MN) | Dry-dispersed | MIT | N/A |
| Illite | IS03 | Illite NX (Arginotec, Germany) | Dry-dispersed | KIT | (Hiranuma et al., 2015a) |
| Fly ash | N/A | Four samples of fly ash from U.S. power plants: J. Robert Welsh Power Plant (Mount Pleasant, TX), Joppa Power Station (Joppa, IL), Clifty Creek Power Plant (Madison, IN) and Miami Fort Generating Station (Miami Fort, OH) (Fly Ash Direct, Cincinnati, OH) | Dry-dispersed | MIT | (Garimella, 2016; Zawadowicz et al., 2016) |
| Na-Feldspar | FS05 | Sodium and calcium-rich feldspar, samples provided by Institute of Applied Geosciences, Technical University of Darmstadt (Germany) and University of Leeds (UK) | Dry-dispersed | KIT | (Peckhaus et al., 2016) |
| K-Feldspar | FS01 | Potassium-rich feldspar, samples provided by Institute of Applied Geosciences, Technical University of Darmstadt (Germany) and University of Leeds (UK) | Dry-dispersed | KIT | (Peckhaus et al., 2016) |
| Agar | N/A | Agar growth medium for bacteria, Pseudomonas Agar Base (CM0559, Oxoid Microbiology Products, Hampshire, UK) | Wet-generated | KIT | N/A |
| Bacteria | PS32B74 + PFCGina01 | Two different cultures of *Pseudomonas syringae*. | Cultures grown on the agar growth medium and wet-generated | KIT | (Zawadowicz et al., 2016) |





| | | | | | |
|---|---|---|---|---|---|
| Cellulose | MCC01, FC01 | Microcrystalline and fibrous cellulose (Sigma Aldrich, St. Louis, MO) | Wet-generated | KIT | (Hiranuma et al., 2015b) |
| Hazelnut | PWW-hazelnut | Natural hazelnut pollen (GREER, Lenoir, NC) wash water | Wet-generated | KIT | (Zawadowicz et al., 2016) |
| Snomax | Snomax | Snomax, (Snomax International, Denver, CO) irradiated, desiccated and ground *Pseudomonas syringae* | Wet-generated | KIT | (Zawadowicz et al., 2016) |
| PSL | N/A | Polystyrene latex spheres (Polysciences, Inc. Warrington, PA), various sizes | Wet-generated | MIT | N/A |
| Soot | CAST minOC or maxOC | CAST soot | miniCAST flame soot generator (manufactured by Jing Ltd Zollikofen, Switzerland) | KIT | (Henning et al., 2012) |
| SOA | SOA | Secondary organic aerosol | Ozonolysis of $\alpha$-pinene | KIT | (Saathoff et al., 2003) |
| K-Feldspar cSA | FS01cSA or FS04cSA | Potassium-rich feldspar (as above) coated with sulfuric acid (SA). | Sulfuric acid incrementally added to the chamber filled with K-feldspar to achieve coatings | KIT | (Saathoff et al., 2003) |
| K-Feldspar cSOA | FS04cSOA | Potassium-rich feldspar (as above) coated with secondary organic aerosol (SOA, as above). | Sulfuric acid incrementally added to the chamber filled with K-feldspar to achieve coatings | KIT | (Saathoff et al., 2003) |

2    Table 1. Description of aerosol types used in training data set.




| Aerosol Type | | | | Broad Categories | | | |
| --- | --- | --- | --- | --- | --- | --- | --- |
| Negative | | Positive | | Negative | | Positive | |
| ion | feature | ion | feature | ion | feature | ion | feature |
| 35 | $^{35}Cl^-$ | 23 | $Na^+$ | 35 | $^{35}Cl^-$ | 23 | $Na^+$ |
| 25 | $C_2H^-$ | 59 | $Co^{+(1)}/CaF^+/C_2H_2OOH^+$ | 26 | $CN^-/C_2H_2^-$ | 59 | $Co^{+(1)}/CaF^+/C_2H_2OOH^+$ |
| 24 | $C_2^-$ | 39 | $^{39}K^+$ | 46 | $NO_2^-$ | 44 | $SiO^+/COO^+/^{44}Ca^+/AlOH^+$ |
| 57 | $C_2OOH^-$ | 12 | $C^+$ | 1 | $H^-$ | 39 | $^{39}K^+$ |
| 59 | $C_2H_2OOH^-/AlO_2^-$ | 24 | $C_2^+$ | 57 | $C_2OOH^-$ | 28 | $Si^+/CO^+$ |
| 43 | $HCN^-/AlO^-$ | 41 | $^{41}K^+/C_3H_5^+$ | 59 | $C_2H_2OOH^-/AlO_2^-$ | 41 | $^{41}K^+/C_3H_5^+$ |
| 1 | $H^-$ | 204-208 | Pb region ($^{204}Pb$, $^{206}Pb$, $^{207}Pb$ and $^{208}Pb$) | 45 | $COOH^-$ | 54 | $^{54}Fe^+$ |
| 26 | $CN^-/C_2H_2^-$ | 27 | $Al^+/C_2H_3^+$ | 42 | $CNO^-/C_2H_2O^-$ | 56 | $Fe^+/CaO^+$ |
| 46 | $NO_2^-$ | 44 | $SiO^+/COO^+/^{44}Ca^+/AlOH^+$ | 43 | $HCN^-/AlO^-$ | 27 | $Al^+/C_2H_3^+$ |
| 16 | $O^-$ | 57 | $^{57}Fe^+/CaOH^+/C_3H_4OH^+$ | 16 | $O^-$ | 45 | $SiOH^+/COOH^+$ |
| 17 | $OH^-$ | N/A | aerodynamic diameter | 73 | $C_2O_3H^-/C_3H_2OOH_3$ | 66 | $Zn^+$ |
| 61 | $SiO_2H^-/^{29}SiO_2^-/C_5H^-/CHO_3^-$ | 83 | $H_3SO_3^-/C_4H_2OOH^+$ | 63 | $PO_2^-$ | 57 | $^{57}Fe^+/CaOH^+/C_3H_4OH^+$ |
| 63 | $PO_2^-$ | 87 | $^{87}Rb^+/CaPO^+$ | 60 | $SiO_2^-/C_5^-/CO_3^-/AlO_2H^-$ | 87 | $^{87}Rb^+/CaPO^+$ |
| 19 | $F^-/H_3O^-$ | 13 | $CH^+$ | 15 | $NH^-/CH_3^-$ | 85 | $^{85}Rb^+$ |
| 76 | $SiO_3^-$ | 66 | $Zn^+$ | 24 | $C_2^-$ | 83 | $H_3SO_3^+/C_4H_2OOH^+$ |
| 77 | $SiO_3H^-/^{29}SiO_3^-$ | 28 | $Si^+/CO^+$ | 76 | $SiO_3^-$ | 24 | $C_2^+$ |
| 79 | $PO_3^-$ | 85 | $^{85}Rb^+$ | 32 | $O_2^-$ | 204-208 | Pb region ($^{204}Pb$, $^{206}Pb$, $^{207}Pb$ and $^{208}Pb$) |
| 60 | $SiO_2^-/C_5^-/CO_3^-/AlO_2H^-$ | 72 | $FeO^+/CaO_2^+$ | N/A | aerodynamic diameter | 40 | $Ca^+$ |
| 45 | $COOH^-$ | 54 | $^{54}Fe^+$ | 71 | $C_3H_2OOH^-$ | 153 | $^{137}BaO^+$ |
| N/A | aerodynamic diameter | 82 | $ZnO^+$ | 50 | $C_4H_2^-$ | N/A | aerodynamic diameter |

$^{(1)}$ Contamination

Table 2. Features rankings for differentiation of particles between labels and between broad categories in positive and negative ion modes. See text for additional details.



| Category | Negative | Postive |
|---|---|---|
| Fertile Soil | 0.89 | 0.84 |
| Mineral/Metallic | 0.93 | 0.97 |
| Biological | 1.00 | 1.00 |
| Other | 0.93 | 0.96 |

| Category | Negative | Postive |
|---|---|---|
| Fertile Soil | $0.022 \pm 0.021$ | $0.032 \pm 0.031$ |
| Mineral/Metallic | $0.017 \pm 0.031$ | $0.006 \pm 0.013$ |
| Biological | 0.000 | $0.001 \pm 0.003$ |
| Other | $0.025 \pm 0.075$ | $0.010 \pm 0.029$ |

Table 3. Model performance by category and ion mode on a population consisting
entirely of aerosols within that category. Left: Average classification accuracy where 1.0
= 100% precision (Powers, 2007). Right: mean and standard deviations of
misclassification.



## 1 Figure

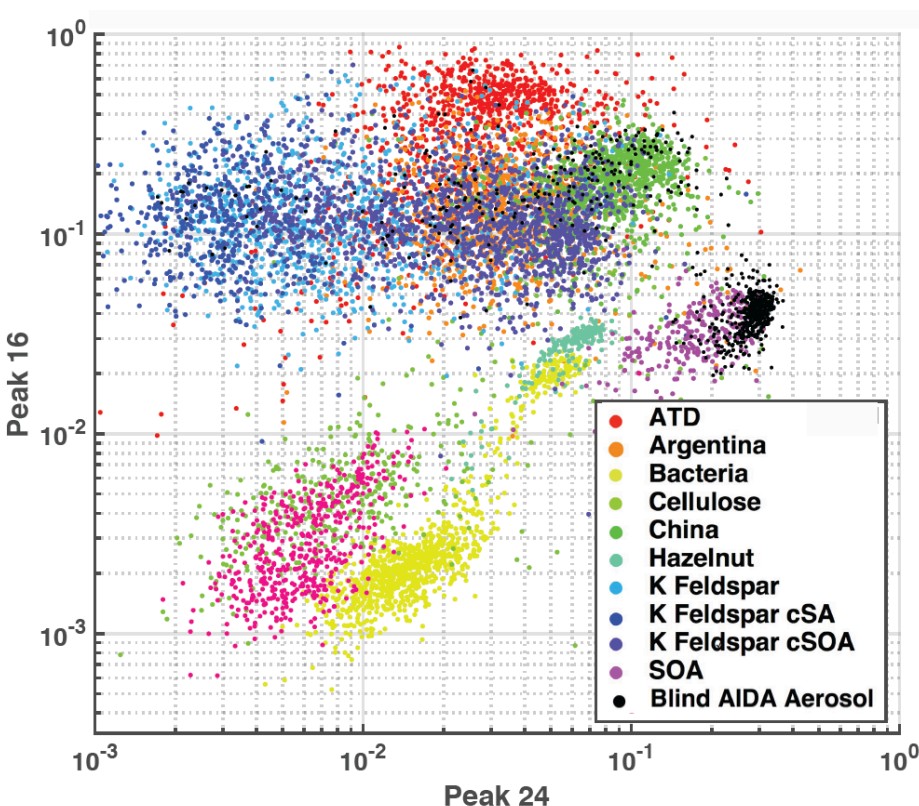

Figure 1: Aerosol training data plotted as feature area 16 (O⁻) verses area 24 (C₂⁻). Axes
represent peak areas normalized to total signal obtained from PALMS (i.e., 1 = 100% of
signal). This illustrates simple 2-dimensional clustering of aerosols from the training data
set by type. Co-plotted are ~500 randomly drawn spectra from the AIDA blind
experiment, which were known to be a subset of the training data aerosols.





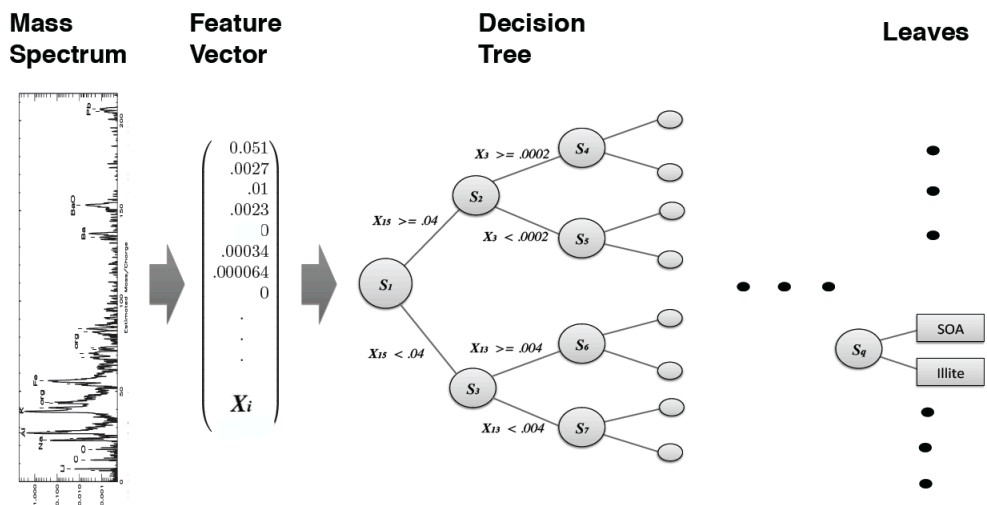

Figure 2. Schematic of decision tree classification for a single aerosol spectrum. From
left to right, a mass spectrum is normalized with respect to total ion current, forming the
elements of normalized feature vector X. A trained decision tree then applies a series of
tests to a discreet number of peaks in order to arrive at a categorical aerosol prediction
(the leaves).





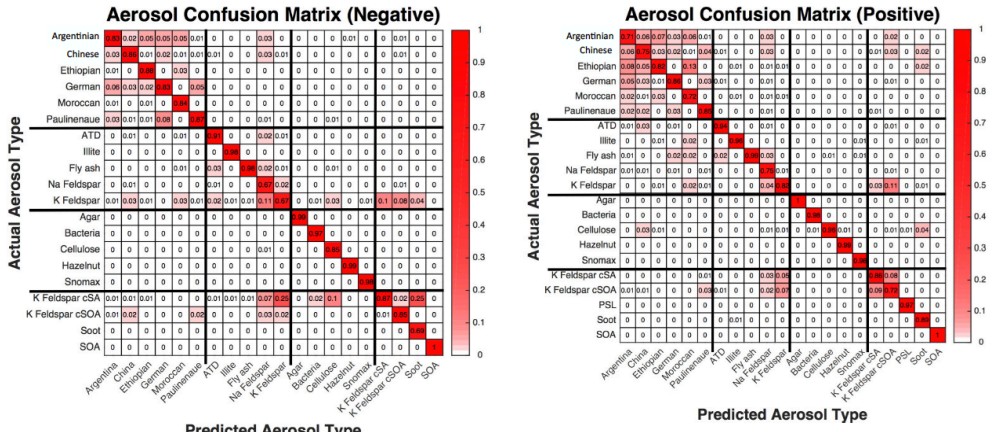

Figure 3. Column-normalized confusion matrices showing fraction of aerosols labeled as
j that belong to i, where i and j are row and column indices, respectively. Confusion
matrices are determined from training data of known origin and are used to compute
probability distributions. Aerosol types (Table 1.) are grouped into four broad categories
delineated by the bold horizontal and vertical bars. From top to bottom or left to right:
fertile soils, mineral/metallic, biological, and other. Classification accuracy, the average
probability of a correct aerosol prediction across all labels, is computed by averaging
diagonal matrix elements. For all aerosol types, the accuracy is 88% in positive ion mode
and  86% in negative ion mode.





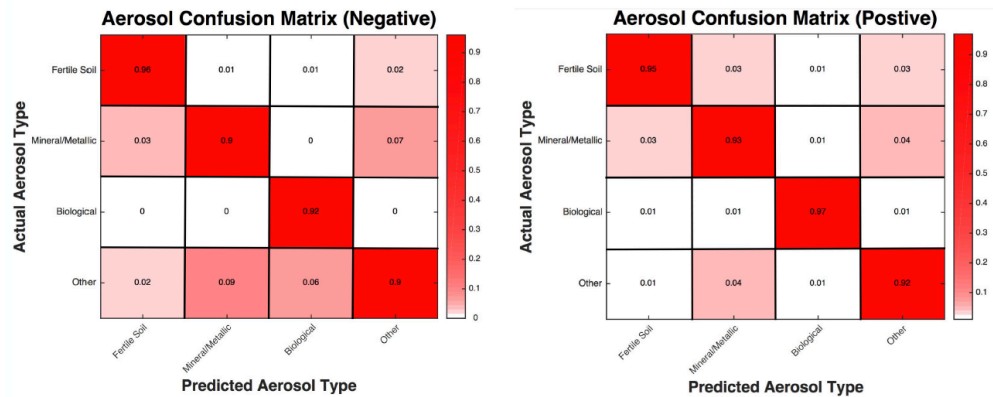

2    Figure 4. Column-normalized confusion matrices for the broad categorization of aerosols

3    following the convention in Figure 3. For all aerosol categories, the accuracy is 94% in

4    positive ion mode and  92% in negative ion mode.





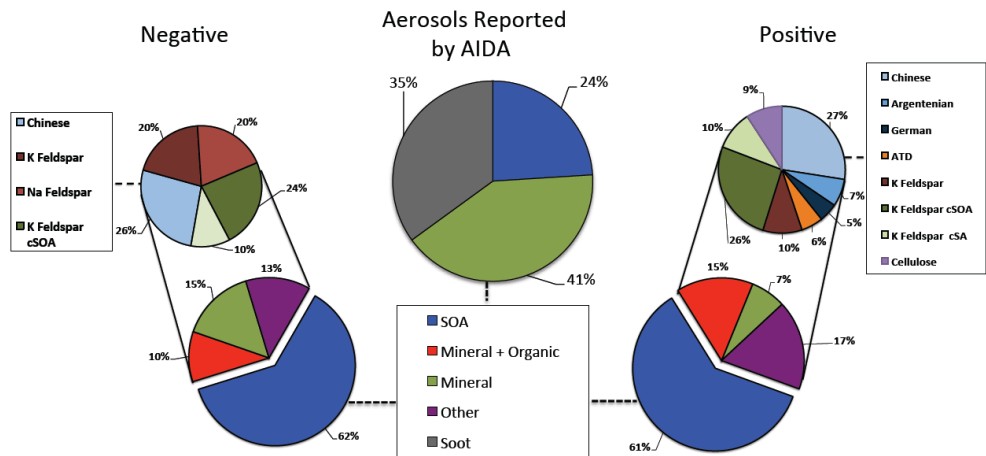

Figure 5. Model predictions of ~5000 aerosols sampled from the AIDA FIN01 blind
mixture which was known to be a subset of the training data. Top middle : aerosol types
input to the chamber for the blind mixture. Model predictions are shown for negative and
positive ion mode on the left and right, respectively. Bottom: broad categories. Top:
breakout by aerosol type of the non-SOA categories above the 1% level. Notes (1) the
soot in the blind mixture was known to be below the instrument detection limit and
therefore is not expected to be found in the data, (2) coagulation of SOA and mineral dust,
which occurred after aerosol input to the chamber, appears as the mineral + organic
category.