# Peer review of "A Machine Learning Approach to Aerosol Classification for Single"

_Atmospheric Measurement Techniques, 2017_

## Referee Comment (RC1) · Anonymous Referee #1 · 15 Mar 2018

The authors apply a machine learning technique (random forests) which is known for its good predicting capabilities to a very interesting and unique SPMS data set. The goal is to predict the composition of an unknown artificial aerosol mixture. Whose constituents are known to belong to a group of aerosol classes that had been analyzed beforehand. The technique is a very promising approach and especially the information extracted from the training-algorithm seems to be very valuable.

But in my opinion several major issues need to be discussed:

-The main scientific work and ideas that were put into the paper are the growing of the forests and their validation including variable reduction and creation of the confusion

matrix. Accordingly, four out of five figures are about these topics. And the results are interesting and offer new ways of looking at this kind of datasets. In contrast the abstract, mainly the introduction and the conclusion strongly focus on the prediction of aerosol classes .

-The paper is in its present form hard to follow. Often the nomenclature is not consistent throughout the paper or doesn't fit to the cited literature.

For example, they do not use the proper term "random forest" but call it machine learning classifier, predictive model, classification model, rule-based probabilistic classification of a decision tree ensemble or supervised classification

and even more important

-While the basic algorithm to grow a random forest is presented. The underlying concepts (randomness, law of big numbers, assumptions, input parameter) and details of the validation process remain unclear.

For example, k=1000 trees have been used for each forest but no further explanation is given why exactly this number of trees is the right one. Or a plot of the test set error against number of trees presented which would make this decision obvious.

The number of random variables used to select the best split from is not specified nor its implications discussed.

The treatment of the "out-of-bag" observations, which is the central means of validation is not comprehensible.

The resultant classification accuracies are not put into perspective; thus the reader can't judge if the algorithm performs is a major improvement to other methods, of which the simplest would be to just use mean values of each aerosol class and use the most similar one as a prediction. It is not given which implementation of the algorithm is useed. Nor how long a typical random forest generation lasts and how this runtime scales with respect to number of particles, number of trees, number of split variables,

etc. . Along with the memory requirements which are missing too, these are basic and easy to provide information that help to compare this method to other methods.

-The random forests have been grown on chemical information and the size of individual aerosol particles, but some of the aerosol classes are not chemically defined. (e.g. multiple fertile soil classes, ATD) This basic contradiction is not clearly addressed.

-To me the section dealing with the blind test data does not fit to the abstract and introduction which present the random forest as a tool specifically suited for this use-case. After showing 80+

So my suggestion would be to split the paper and resubmit both parts in a thoroughly revised version one part with a clear focus on the algorithm and its general applicability to SPMS data including a real comparison to methods currently used (fuzzy-cmeans, manual decision tree, k-means).

And the other with a thorough analysis of the blind data set explaining in a comprehensible way the measured spectra based on all available information, statistics and assumptions. If it is not possible to explain the measured spectra in a controlled laboratory experiment like the one described, the use of the instrument to characterize atmospheric aerosol populations would be quite limited.

---

## Referee Comment (RC2) · Anonymous Referee #2 · 11 Apr 2018

Overall, this paper strives to explain a particular application of a machine learning approach to the classification of single particle mass spectra and to describe the performance of the approach to a particular "blind" test. The dataset that is used to train and test the machine learning approach is a good one for such a study, containing a variety of different particle types. The major shortfall of this paper is that the authors neither explain the details of the machine learning approach fully nor do they fully engage with the aerosol classification results, leaving the reader somewhat confused in both realms. In addition, the authors do not attempt to address the performance of their approach in terms of time, or give information about how applicable it would be to ambient data sets where particles would not necessarily be of such distinct types.

[Figure]

Finally, they given no metrics for success – how good is good enough performance for this approach? How good are other methods, compared to that presented here?

I would recommend that this paper be significantly revised, in such a way that a) the machine learning approach can be fully described and choices made justified with data, and b) the aerosol particle classification results can be fully examined and compared to other methods.

Specific Comments: 1. The paper reads as if it was written by two separate people, one for the algorithm discussion and one for the aerosol particle classification discussion. This should be addressed as a final version (or versions) is developed. For example, on p. 4, the transition between lines 12 and 13 is abrupt and jarring. 2. The authors refer to "volatile" components of aerosol particles multiple times in the paper (first p. 3, line 13). I believe they mean semi-volatile, or at least "more volatile" than other components. Volatile species would not be expected to be found in particles. 3. In section 2.2, where the training data set is introduced, the authors need to discuss the applicability of this dataset to any "real" experiment. Would these particles be a good representation of ambient particles, for example? 4. In the discussion of the data presented in Table 2, the authors state that the columns labeled "broad" are applicable to the categorization of the particles when they are lumped together into broad chemically-similar categories. These categories should be defined in this context (and not just in the context of the confusion matrices), presumably in Table 1. Any interpretation of the differences should be discussed in the results section. In addition, the paragraph on p. 11 about the 59+ ion observed in some samples (which ones?) should be moved into the results section. Finally, is it certain that 59+ is Co rather than an organic contaminant? 5. The authors provide no information about the average mass spectra of the individual particle types and the variability within "identical" particles or between particle types. This would seem to be an important parameter in determining how well the algorithm can do to separate them. Based on the two peaks shown in Figure 1, this is an important factor. 6. The discussion of confusion matrices was confusing. Essentially, these matrices

represent normalized counts of the sorting of known particles into the available classes. This can be stated much more cleanly than the three page description provided on pp. 12 – 14. This is another example of a section that is trying to be both an algorithm and a particle chemistry paper, and not mixing the two effectively. 7. Figure 5 and the discussion of the "blind" test in Section 3.2 are key to the goals of this paper, but are confusing in their presentation. Regarding the text, why do the authors not know the number of particle types that were used in the challenge (p. 15, line 2 "3 -4 aerosol types . . . were aerosolized")? How well can the results be evaluated if the test conditions are not known? The authors describe a probabilistic correction for the mis-labeling that they observe in their confusion matrices (p. 15, lines 14 – 18), and say that the results "better represent the underlying aerosol population." (p. 15, lines 17 – 18), but they don't provide the data to evaluate this claim. The data presented in Figure 5 do not make the case that the authors are trying to make. While the two models (positive and negative) show relatively good agreement with each other, the representation of the particles introduced into the chamber is poor. The authors show the breakdown of soot, SOA, and mineral particles introduced in Figure 5, state that the soot particles are too small to see with their instrument, and then compare against the soot-containing dataset anyway. If the pie that represents "Aerosols Reported by AIDA" were renormalized to include only observable particles in this experiment, SOA would represent 44% of the pie and mineral would represent 56% – assuming that the "Aerosols Reported by AIDA" pie is also representing number of particles, rather than mass of introduced particles (this is not stated). If this pie represents something other than number, there is no comparison to the blind sample possible in this figure, only a comparison between the two models.

Technical Corrections: 1. P. 10, line 17: remove the word "rows" from the line. 2. P. 15, line 3: "AIDA" is written as "ADIA." 3. Figure 1 is very difficult to read. The black points for the "Blind AIDA Aerosol" are only visible on the right-most part of the graph, in the region of (0.3, 0.04) and the similar colors are hard to differentiate. Consider a figure like this that is broken out into the broad categories of particle types. 4. Figure

5, negative model, the small pie includes 5 wedges but only 4 labels.

---

## Author Comment (AC1) · 21 Jun 2018

We would like to thank the reviewers for their comments. We apologize for the extended time between reviews and revision but this was due to incorporating the important extra work requested in the revised manuscript. We have made the changes suggested and responded to comments in a point by point format below. We believe this is a stronger paper as a result and thank the reviewers for their work.

**Reviewer #1**

1.) The main scientific work and ideas that were put into the paper are the growing of the forests and their validation including variable reduction and creation of the confusion matrix. Accordingly, four out of five figures are about these topics. And the results are interesting and offer new ways of looking at this kind of datasets. In contrast the abstract, mainly the introduction and the conclusion strongly focus on the prediction of aerosol classes.

In response to this reviewer comment we have added text in the three stated areas (abstract, intro and conclusions) to the emphasize the use of the general technique as opposed to the specific classification of aerosol spectra. For example, we now lead the introduction with the generalized technique and then move to the specific use here. We believe it is also critical to maintain the classification objective as this was our motivation for the work and the data we present.

2: 8-12 in the abstract now reads:
*"Our primary focus surrounds the growing of random forests using feature selection to reduce dimensionality, and the evaluation of trained models with confusion matrices. In addition to classifying 20 unique but chemically-similar aerosol types, models were also created to differentiate aerosol within four broader categories: fertile soils, mineral/metallic particles, biological, and all other aerosols."*

3: 1-15 in the introduction:
*"Following the introduction of random forests in the 1990s, recent developments in deep learning and neural networks have triggered a renewed interest in machine learning. This has led to the development of numerous easy-to-use, freely-available, open-source packages in popular programming languages like Python, and these tools are becoming increasing used in academia and industry. While random forests have been used for complex classification and regression analysis in various fields, studies that employ random forests in aerosol mass spectrometry remain sparse. Utilizing these tools, the primary purpose of our study is to introduce a framework for growing random forests, reducing dimensionality, ranking chemical features, and evaluating performance using confusion matrices. Such properties are desirable for SPMS studies, where input variables can become redundant and interpretability is more limited methods with methods such as cluster analysis and neural networks. Powerful analysis techniques such as those falling out of recent artificial intelligence research can prove useful for*

*helping to tease out the subtle yet significant impact that aerosol chemistry has on the climate system."*

19: 16-21 in the conclusion:
"This study lays out a framework for training and implementing random forests on SPMS data, with a focus on dimensionality reduction and the evaluation of model performance with confusion matrices. A key benefit to the proposed method is chemical feature selection, which allows researchers to identify potentially important chemical markers between arbitrary groups of aerosols or identify sources of contamination. Additionally, the approach allows for differentiation of aerosols within a SPMS dataset, augmenting existing tools and reducing the need for a qualitative comparison between mass spectra."

2.) The paper is in its present form hard to follow. Often the nomenclature is not consistent throughout the paper or doesn't fit to the cited literature. For example, they do not use the proper term "random forest" but call it machine learning classifier, predictive model, classification model, rule-based probabilistic classification of a decision tree ensemble or supervised classification and even more important.

We thank the reviewer for the suggestion and agree more specific language should be applied to describe the random forest algorithm, which is the primary focus of the paper. The terminology through the paper has been modified to be consistent and precise per the comments, and we have removed sections that went into unnecessary detail.
While do note the terms mentioned are appropriate for introducing and motivating the random forest approach, they describe broader categories of machine learning models. The distinction between each was not necessary for this paper, beyond the mention of how supervised methods (i.e. classification via random forests) differ from unsupervised methods such as clustering. We have now tried to make this distinction and believe it is now more clear.

3.) While the basic algorithm to grow a random forest is presented. The underlying concepts (randomness, law of big numbers, assumptions, input parameter) and details of the validation process remain unclear...

As suggested, we have modified the text in several ways to address this comment, which has been broken into specific points and addressed:

Upon further inspection, the authors agree the concepts you mention such as randomness and out-of-bag sampling are key details surrounding random forests approach and needed further detail. Out-of-bag samples refers to spectra that are held out during the validation process to prevent training and testing on the same data, and the section has been revised to more concisely explain the procedure.

12: 2-10 now reads:

*"Overall, the generalizability and robust performance of random forests is owed significantly to the series of random statistical procedures used to construct such models. An ensemble classifier reduces variability by averaging predictions over a series of independently trained models, and bagging introduces additional randomness by producing "perturbed" versions of the original data via random sampling of input data. The randomness used in constructing forests, both in bagging the training set and choosing variable splits, work to decorrelate the output of each tree even as the inputs become correlated [Breiman, 2001]. As the number of trees increases, the law of large numbers guarantees a convergence of the out-of-bag error to the generalization error."*

…For example, k=1000 trees have been used for each forest but no further explanation is given why exactly this number of trees is the right one. Or a plot of the test set error against number of trees presented which would make this decision obvious. The number of random variables used to select the best split from is not specified nor its implications discussed….

- We have included various pieces of the information you requested in the paper, and agree the details surrounding runtime, memory requirements, and model selection are important. Additionally, supplementary plots that characterize how model performance and runtime scale are provided below. The plot also shows how runtime scales with number of trees and number of split variables.

11:12 - 20 now includes the requested information:
*"The number of variables per split is chosen to be 11 and the number of trees is 110. The optimal model was determined by enumerating combinations of these parameters on a coarse grid and selecting the values that produce the lowest test error. Model behavior is primarily sensitive to the number of variables per split, and shows weak dependence on the number of trees and number of input variables beyond small values. As the number of variable splits increases, error decreases exponentially to a local minimum before again rising due to over fitting. Alternatively, as the number of trees is increased the error asymptotes to some nonzero value, a known characteristic of random forests where test error converges to the generalization error.*

[Figure]

…The treatment of the "out-of-bag" observations, which is the central means of validation is not comprehensible….

More detail has been added.

10:15 - 10:22 now reads:

*"On average, each tree is built with ~63% of the original data, leaving a portion of the training set unsampled. The unsampled data for each tree, known as 'out-of-bag' observations, are recorded and later provide a means to assess classification error for the forest. To determine model error, predictions are made on each point in the dataset using only the subset of trees that did not use the point for training. Each training point is left out at least once. This is analogous to making predictions with a separately trained forest that did not observe the point and prevents testing with the same data used for training."*

…The resultant classification accuracies are not put into perspective; thus the reader can't judge if the algorithm performs is a major improvement to other methods, of which the simplest would be to just use mean values of each aerosol class and use the most similar one as a prediction…

We have compared the technique to a simple classifier that uses the euclidean distance to assigned an unknown aerosol to the closest "mean" class vector. Confusion matrices for the broad categories have been included in the paper for comparison as part of figure 4, and matrices for all labels have been included below as a supplementary figure.

13:10 - 20 now reads:

*"To access relative model performance, we contrast the results with a simple classifier that compares unseen aerosols to a set of class mean vectors. Using the Euclidean distance metric, the unknown aerosol is assigned to the nearest class. This simple baseline classifier helps put results in the context of machine learning techniques that rely on distance-based metrics such as k-means and hierarchical clustering. K-means clustering attempts to divide the data points into k distinct clusters, representing spectra as vectors. Using Euclidean distance, the standard algorithm assigns points to centroids, or clusters, which are essentially mean vectors representing the average of all points in the cluster. Assuming perfect convergence of k-means clustering, where k is the number of aerosol classes, each cluster represents the mean of aerosol in that class. The random forest results below demonstrate many areas of improvement over the simple classifier."*

[Figure]

…It is not given which implementation of the algorithm is used. Nor how long a typical random forest generation lasts and how this runtime scales with respect to number of particles, number of trees, number of split variables, etc. . Along with the memory requirements which are missing too, these are basic and easy to provide information that help to compare this method to other methods….

*11:20 – 12:1*
*"The models were trained with the Python 2.7 Scikit-learn module on a MacBook Pro with 16 GB 1600 MHz DDR3 memory and a 2.5 GHz Intel Core i7 processor. A typical*

*random forest model took about 5-10 seconds to train, and we found a linear relationship between runtime and both the number of trees and variables per split."*

4.) The random forests have been grown on chemical information and the size of individual aerosol particles, but some of the aerosol classes are not chemically defined. (e.g. multiple fertile soil classes, ATD) This basic contradiction is not clearly addressed.

We agree with the reviewer and have expanded the paper to define this more clearly. This is in the form of the new paragraph in the introduction on page 5 which also includes detail that some of this issue stems from the complex nature of atmospheric aerosols that are often defined by source as opposed to type. Furthermore, we have color coded Table 1 to make the distinction more clear.

> 4:16 – 5:9
>> "Chemical composition of an individual atmospheric aerosol particle is a complex interplay between its primary composition at the source (i.e. dust, biogenic organic, anthropogenic organic, soot, etc.) and its atmospheric processing up to the time of detection. Atmospheric processing can include any combination of coating with secondary material, coagulation and cloud processing. Even distinct primary aerosol types can have similar mass spectral markers. For example, fly ash, mineral dust and bioaerosol can all contain strong phosphate signal [Zawadowicz et al., 2017]. Secondary material is often difficult to differentiate from primary material, but even minor compositional changes can be atmospherically important. As one example, mineral dusts are known to be effective at nucleating ice clouds [Cziczo et al., 2013]; however, despite minor addition of mass, atmospherically processed mineral dust is less suitable for ice formation [Cziczo et al., 2013]. As a second example, ice nucleation in mixed-phase clouds has been suggested to be predominantly influenced by feldspar, a single component among the diverse mineralogy of atmospheric dust [Atkinson et al., 2013]. Using current SPMS data analysis approaches, it can be difficult to detect these minor yet important compositional differences and new robust and generalizable analysis techniques are critical."

5.) To me the section dealing with the blind test data does not fit to the abstract and introduction which present the random forest as a tool specifically suited for this use-case. After showing 80+

There are two primary factors that help explain differences between the test set and blind set, which are both due to the way the experiment and sampling were conducted : a) transmission efficiency b) coagulation. During the course of the experiment, we expect the mineral dust and SOA to coagulate. Since aerosol types were reported by AIDA before particles enter the chamber, it is not possible to quantify exactly what fraction of the particles picked up an SOA coating. Moreover, there would have been a time dependence to the coagulation process.

Additionally, through coagulation, there is the possibility of effectively producing a particle type not in the training set, depending on the exact mineral component of the mineral dust used by AIDA. While it is known a mineral dust was included in the chamber, the exact composition of the dust was not known. While our training set contains K-Feldspar coated with SOA, a different type of SOA-coated mineral dust will appear unique to the model. Because the generalization performance of supervised classifiers is ill-defined for particles not included in the training set, this could lead to performance that is not captured by the confusion matrices. Given the experimental uncertainties from transmission efficiency and coagulation, as well as the model uncertainties highlighted in the confusion matrices, we believe the results reveal skill in using random forests to pick out distinct aerosols. In future studies, uncertainties can be reduced by adding additional particle labels or accounting for transmission efficiency, but coagulation will likely remain an inherent uncertainty. The limitations of transmission efficiency and coagulation are also noted at the end of the results section.

The caption in Figure 5 has been updated to state these factors more clearly
            35: 6-10 now reads
    "Notes (1) the soot in the blind mixture was known to be below the instrument detection limit and therefore is not expected to be found in the data, (2) coagulation of SOA and mineral dust, which occurred after aerosol input to the chamber, appears as the "other" category, (3) the aerosols types reported by AIDA do not account for PALMS transmission efficiency."

So my suggestion would be to split the paper and resubmit both parts in a thoroughly revised version one part with a clear focus on the algorithm and its general applicability to SPMS data including a real comparison to methods currently used (fuzzy-cmeans, manual decision tree, k-means). And the other with a thorough analysis of the blind data set explaining in a comprehensible way the measured spectra based on all available information, statistics and assumptions. If it is not possible to explain the measured spectra in a controlled laboratory experiment like the one described, the use of the instrument to characterize atmospheric aerosol populations would be quite limited.

The co-authors had a discussion regarding this suggestion but have decided that keeping the paper in something similar to the original format was the best course of action. Our reason is that the current format allows us to present a new technique for aerosol mass spectra classification and then use it on a single well constrained data set. We expect to expand its usage on future (e.g. field) data sets.

We acknowledge the reviewer's comments have focused on the general method and they raise an interesting point about using multiple method on a data set and then doing a cross comparison. We need to therefore comment that, in keeping this a new method to use with a demonstration on this data set, a multi-method comparison is well beyond the scope of our goal. The reviewer notes several different methods that have been used for aerosol mass spectral analysis. However, in practice, each instrumental group has focused on one or two techniques.

Setting up a means to apply several techniques to a single data set is therefore non-trivial and would occupy a significant amount of time. Again, we agree the cross comparison is interesting but would certainly represent months of research time beyond the scope of what we are attempting to do here.

---

## Author Comment (AC2) · 21 Jun 2018

We would like to thank the reviewers for their comments. We apologize for the extended time between reviews and revision but this was due to incorporating the important extra work requested in the revised manuscript. We have made the changes suggested and responded to comments in a point by point format below. We believe this is a stronger paper as a result and thank the reviewers for their work.

**Reviewer #2**

As an overview comment we repeat the introductory review text here. Each point made is repeated specifically addressed below.

The major shortfall of this paper is that the authors neither explain the details of the machine learning approach fully nor do they fully engage with the aerosol classification results, leaving the reader somewhat confused in both realms.

In addition, the authors do not attempt to address the performance of their approach in terms of time or give information about how applicable it would be to ambient data sets where particles would not necessarily be of such distinct types.

Finally, they given no metrics for success – how good is good enough performance for this approach? How good are other methods, compared to that presented here?
I would recommend that this paper be significantly revised, in such a way that a) the machine learning approach can be fully described and choices made justified with data, and b) the aerosol particle classification results can be fully examined and compared to other methods.

Specific Comments:
1. The paper reads as if it was written by two separate people, one for the algorithm discussion and one for the aerosol particle classification discussion. This should be addressed as a final version (or versions) is developed. For example, on p. 4, the transition between lines 12 and 13 is abrupt and jarring.

We regret the paper seemed disjoint and believe this is partially from the fact the we are using a new technique on a more traditional dataset. It was not, as the reviewer suggested, written by two different people and knit together. We have gone through and tried to streamline the flow of the paper by removing the less pertinent details surrounding the algorithm, including your suggestion of simplifying the discussion of confusion matrices – please see the full track changes version. We are attempting to characterize two distinct topics as they related to the paper: The details surrounding training and applying a random forest as well as aerosol populations in the content of mass spectrometry.

To directly address this example, 4:12-17 now reads:

"To pick up on these minor yet important compositional differences, robust and generalizable analysis techniques are critical. We show that supervised training with random forests can differentiate aerosols in SPMS data more accurately than simpler approaches."

2. The authors refer to "volatile" components of aerosol particles multiple times in the paper (first p. 3, line 13). I believe they mean semi-volatile, or at least "more volatile" than other components. Volatile species would not be expected to be found in particles.

The reviewer is correct and we have replaced the term volatile with semi-volatile.

3. In section 2.2, where the training data set is introduced, the authors need to discuss the applicability of this dataset to any "real" experiment. Would these particles be a good representation of ambient particles, for example?

We have addressed your question by including the following.
6: 6-11

"The choice of supervised or unsupervised machine learning will depend on the researcher's use-case, and each method has unique advantages and disadvantages. We note a limitation of the random forest approach - and for supervised learning in general - is the inability to classify aerosol types outside of the training set. The ability of a random forest to characterize ambient atmospheric datasets, therefore, will strongly depend on which aerosols are contained within the training set."

Although it is feasible that unseen aerosol types will be assigned to the most chemically-similar label, supervised models are tuned to only make predictions on labels in the training set. The error statistics cannot be fully quantified for datasets with unknown aerosol types, so the model may not conform to the determined generalization error. In general, more particle types lead to a more generalizable classifier with better quantifiable error statistics. A study looking chiefly at atmospheric spectra would benefit from adding additional aerosol types and augmenting the analysis with existing methods such as clustering, which are designed to handle unlabeled data.

4. In the discussion of the data presented in Table 2, the authors state that the columns labeled "broad" are applicable to the categorization of the particles when they are lumped together into broad chemically-similar categories. These categories should be defined in this context (and not just in the context of the confusion matrices), presumably in Table 1. Any interpretation of the differences should be discussed in the results section. In addition, the paragraph on p. 11 about the 59+ ion observed in some samples (which ones?) should be moved into the results section. Finally, is it certain that 59+ is Co rather than an organic contaminant?

In this case, the contaminant is almost certainly $Co^+$ originating from tungsten carbide grinder used to process some of the dust. A typical spectrum that shows the nature of the contaminant is shown below. The spectrum has markers that correspond to $Co^+$, $W^{+2}$ and $W^+$ but no obvious organic markers. Alternate assignments for m/z +59 are possible (and we present them in Table

2), but a prominent m/z +59 peak in this dataset is always associated with tungsten carbide contamination shown below.

[Figure]

Table 1 has been color-coded to make the broad category definitions more clear. Additionally we move the paragraph into the discussion as requested and expand:

16:22 – 17:5
"It is noteworthy that while most of the features are logical differentiators of the aerosol types investigated in FIN01 there were also surprises. One example is 59+ (cobalt), determined to be one of the most important features for differentiation. Further investigation determined this material was associated with tungsten carbide contaminant from dry powder dispersion equipment used on some samples. The contamination affected feldspar samples used during the second half of the AIDA measurements in particular."

 5. The authors provide no information about the average mass spectra of the individual particle types and the variability within "identical" particles or between particle types. This would seem to be an important parameter in determining how well the algorithm can do to separate them. Based on the two peaks shown in Figure 1, this is an important factor.

To address your point, we have compared the method to a simple classifier which assigns unknown aerosols to the nearest class mean vector using the Euclidean distance metric. This answers your previous question of "how good is good enough" and provides a baseline that directly depends on the distance of aerosol mean vectors in feature space and variability of individual aerosols within each class.  Figure 4 is updated to show the results of such a classifier. A couple of paragraphs have been included to summarize these results.
13:10 - 20 now reads:

*"To access relative model performance, we contrast the results with a simple classifier that compares unseen aerosols to a set of class mean vectors. Using the Euclidean distance metric, the unknown aerosol is assigned to the nearest class. This simple baseline classifier helps put results in the context of machine learning techniques that rely on distance-based metrics such as k-means and hierarchical clustering. K-means clustering attempts to divide the data points into k distinct clusters, representing spectra as vectors. Using Euclidean distance, the standard algorithm assigns points to centroids, or clusters, which are essentially mean vectors representing the average of all points in the cluster. Assuming perfect convergence of k-means clustering, where k is the number of aerosol classes, each cluster represents the mean of aerosol in that class. The random forest results below demonstrate many areas of improvement over the simple classifier."*

16:8 - 20 now reads

*"Across all categories, the random forest shows improvements over the Euclidean classifier in terms of both accuracy and precision. Figure 4 directly compares confusion matrices for the two methods, revealing overall accuracy improvements of at least 20%. The largest improvements are in the fertile soil and other category, where accuracy rises between 20% and 39% with the random forest. Computing the full confusion matrix for the Euclidean technique (as in figure 3) reveals similar results, with far more frequent mislabeling between fertile soils as well as coated/uncoated particles than our approach. These results reinforce the fact that chemically-similar aerosols which overlap in feature space will often be grouped together when using a single, distance-based classifier. The improvement from random forests is likely a result of a) the ensemble approach, which is known to produce better generalizability than single classifiers and b) the tendency of aerosols with similar chemical properties and atmospheric effect to appear mathematically distinct with a distance metric."*

6. The discussion of confusion matrices was confusing. Essentially, these matrices represent normalized counts of the sorting of known particles into the available classes.
This can be stated much more cleanly than the three page description provided on pp.
12 – 14. This is another example of a section that is trying to be both an algorithm
and a particle chemistry paper, and not mixing the two effectively.

The section on confusion matrices has heavy revised and simplified to focus on aspects of the matrix that are directly used for the paper.

14: 2-4 reads:

*"A confusion matrix captures misclassification tendencies by pair-wise matching the model prediction with the true aerosol type or broad category [Powers, 2007], and can be understood as a contingency table matching model predictions to true labels."*

7.  Figure 5 and the discussion of the "blind" test in Section 3.2 are key to the goals of this paper, but are confusing in their presentation. Regarding the text, why do the authors not know the number of particle types that were used in the challenge (p. 15, line 2 "3-4 aerosol types . .

.were aerosolized")? How well can the results be evaluated if the test conditions are not known?...

The purpose of the blind experiment was to determine the capability of each mass spectrometer to determine the number of particle types and their composition; a situation deemed similar to the challenge of atmospheric sampling. This is now explicitly stated in the text at 17:10 "As part of the FIN01 workshop, it was known that an unknown number of aerosol types from Table 1 were aerosolized into the ADIA chamber at unknown size and relative concentration." We realize the wording in the original sentence implied an unknown test but it was meant to indicate the participants were not aware.

b.) The authors describe a probabilistic correction for the mis-labeling that they observe in their confusion matrices (p. 15, lines 14 – 18), and say that the results "better represent the underlying aerosol population." (p. 15, lines 17 – 18), but they don't provide the data to evaluate this claim.

- The proposed probabilistic correction leads to insignificant changes in the final predictions as the computed fractional difference are small relative a) misclassifications between labels and b) uncertainties from having an unseen label. Furthermore, there is no guarantee the blind dataset will conform to the same mislabeling tendencies, as you have mentioned. We have removed the description from the paper and reapplied the method without the correction.

c.) The data presented in Figure 5 do not make the case that the authors are trying to make. While the two models (positive and negative) show relatively good agreement with each other, the representation of the particles introduced into the chamber is poor. The authors show the breakdown of soot, SOA, and mineral particles introduced in Figure 5, state that the soot particles are too small to see with their instrument, and then compare against the soot-containing dataset anyway. If the pie that represents "Aerosols Reported by AIDA" were renormalized to include only observable particles in this experiment, SOA would represent 44% of the pie and mineral would represent 56% – assuming that the "Aerosols Reported by AIDA" pie is also representing number of particles, rather than mass of introduced particles (this is not stated). If this pie represents something other than number, there is no comparison to the blind sample possible in this figure, only a comparison between the two models.

- The aerosols reported by AIDA are number concentrations, but there are several considerations to account for when considering the labels reported by our classifier in the blind dataset.
    - The aerosols reported by AIDA do not account for PALMS transmission efficiency, which depends on the size and aerodynamic properties of aerosols. For example, particles larger than 1000nm are over-reported by the classifier due to increased PALMS efficiency in that range. We now note this inherent limitation.

○ During the course of the experiment, we expect (and observed) the mineral dust and SOA to coagulate. Since aerosol types were reported by AIDA before particles enter the chamber, it is not possible to quantify exactly what fraction of the particles picked up an SOA coating. Additionally there is the possibility of effectively producing a particle type not in the training set, depending on the exact mineralogy of the mineral dust used by AIDA. While it is known a mineral dust was included in the chamber, the exact composition of the dust is not know. The mineral may either contain a specific component or a soil dust. While our training set contains K-Feldspar coated with SOA, a different type of SOA-coated mineral dust will appear unique to the model. Because the generalization performance of supervised classifiers is ill-defined for particles not included in the training set, this could lead to performance that is not captured by the confusion matrices. Given the experimental uncertainties from transmission efficiency and coagulation, as well as the model uncertainties highlighted in the confusion matrices, we believe the results reveal skill in using random forests to pick out distinct aerosols. In future studies, uncertainties can be reduced by adding additional particle labels or accounting for transmission efficiency, but coagulation will likely remain an inherent uncertainty.

These are stated at the end of our results section as well as the caption for Figure 5. The caption in Figure 5 has been updated to mention transmission efficiency as an uncertainty.

35: 6-10 now reads
"Notes (1) the soot in the blind mixture was known to be below the instrument detection limit and therefore is not expected to be found in the data, (2) coagulation of SOA and mineral dust, which occurred after aerosol input to the chamber, was often categorized as mixed mineral and organic particles or fertile soils (i.e., mixtures of mineral and organic components) considered in the training data set, (3) the aerosols types reported by AIDA do not account for PALMS transmission efficiency (see text for details). "

**Technical Corrections:**
1. P. 10, line 17: remove the word "rows" from the line.
   This has been corrected.
2. P.15, line 3: "AIDA" is written as "ADIA."
   This has been corrected.
3. Figure 1 is very difficult to read. The black points for the "Blind AIDA Aerosol" are only visible on the right-most part of the graph, in the region of (0.3, 0.04) and the similar colors are hard to differentiate. Consider a figure like this that is broken out into the broad categories of particle types.
As suggested, we created a similar scatter plot using only broad categories. We still find it difficult to pick out each of the particle types in the region in (0.3, 0.4), even when plotting only small subsets of the training set. This is a result of significant overlapping of aerosol types in the region, and difficult to alleviate. Nevertheless, the aim of the figure is to demonstrate that some types overlap significantly, while others such as SOA form distinct clusters. Since both versions

of the plot demonstrate this to a similar degree, we have decided to leave the original figure in the paper.

[Figure]

4. Figure 5, negative model, the small pie includes 5 wedges but only 4 labels.

This has been corrected.

---

## Author Response (AR2)

We would like to thank both reviewers for their 2nd review of the manuscript and believe it is again stronger for their suggestions. We have addressed each point below and revised the manuscript accordingly.

Reviewer 1

**General Comments:**

The revision has produced a much stronger and more integrated manuscript which describes a particular machine-learning approach used to separate single particle mass spectra by identity. The authors have provided more detailed information about the conceptual framework for their classification scheme, have done a rudimentary comparison to an alternative method, and have done a thorough job of exploring, presenting, and explaining the results from training and "blind" tests using their proposed method.

Although it is mentioned briefly in the manuscript, the authors haven't seriously engaged with assessing the utility of this method for analysis of ambient particle spectra, where presumably it would need to be functional to be useful. Are there situations wherein this method could be used to essentially "pick out" the particles that match one of the training sets, while not trying to differentiate "other" particles not included? If so, how different would the particle spectra need to be to achieve this?

The reviewer correctly suggests a future application of this approach. A paragraph has therefore been added in "Conclusion and Future work" describing an approaching for dealing with ambient datasets that contain aerosol absent from the dataset (21:18 - 22:8).

"For future studies tackling ambient atmospheric data that may contain aerosol types absent from the training set, a form of subspace selection may be used to improve results. The region of parameter space where training data is available can be characterized with a joint probability density function. One such approach is kernel density estimation - a machine learning method that approximates a multidimensional probability density function in a non-parametric manner based on data density. To obtain accurate probability estimates, the method should be fit with a smaller set of important but uncorrelated peaks. The task of classification is then preceded by a filtering step. Spectra residing in the subspace containing the training data should first be identified based on the probability density function. Then, only these particles that are most certain to lie in the training subspace are classified using the classification model as described in this paper. An alternative is to combine the method with clustering by classifying particles in each automatically identified cluster."

Specific Comments:

The authors state, p. 3, lines 11 - 12, that "interpretability is more limited with methods such as cluster analysis and neural networks" without justification. Such statements should include explanations and/or citations, or be removed if they represent opinions.

We have clarified this point to only draw the comparison with neural networks and elaborated. 3: 12 -15 now reads:

"Neural networks rely on a series of variable transformations rectified by nonlinear activation functions, making details of a given classification notoriously difficult to follow. The interpretably and explainability of these models remains an active area of research."

On p. 5, line 12, the authors describe that "algorithms are known to struggle with chemicallysimilar aerosols..." but again provide no definition of "struggle" nor a discussion of how similar is too similar...

The wording has been modified to clarify that chemically-similar particles are often combined together (5:15). The sentence regarding the need to manually combine clusters has also been moved up to better contextualize the difficulties encountered in previous studies (5:16-18). It is noted that an example involving fertile soils was also stated later in the paragraph.

...Furthermore, the discussion on this page, lines 19 - 23, should mention that (as with all of the algorithms discussed in this paper), there are user defined settings that are included in each method, and the choice of those settings influences the outcome significantly. Generalizations about performance are therefore challenging, when little information about settings is provided. An alternative approach that the authors could explore is referencing specific articles in which specific methods/algorithms are used, and commenting on the successes and challenges that are illustrated by the specific results that the authors obtained.

As suggested, the dependence on user-defined settings has been mentioned on 6:18-19. "Additionally, it is noted that comparisons between all machine learning models are sensitive to user-defined parameters and algorithm implementation."

A further response to your mention of user-defined setting is provided in the comment below about Figure 4.

On p. 6, lines 4 - 5, the authors mention "measurement uncertainty" without defining the variable in which that uncertainty is found. Is it the identification, the peak areas, or something else?

We agree these uncertainties should be clarified, so several uncertainties have been explicitly listed in this portion of the paper.

The following has been added on 6: 8-12:

"Uncertainties associated with mass spectrometry include the determination of mass peak areas, internal mixing of aerosols during the experiment, and transmission efficiency. Additionally, the classification method itself introduces and quantifies uncertainty in aerosol identification as a result of imperfect classes separation and parameter uncertainty. In section 2.3, the authors discuss binary decision trees without mentioning random forests, although the term has been introduced. It would be helpful to contextualize the binary trees within the discussion of the random forests at the beginning of this discussion, which could be accompanied by a short comment that the random forest approach will be described more thoroughly below.

We agree context is needed earlier, and have added this on 9: 17 – 19: "A random forest is an ensemble of perturbed decision trees, whereby a final classification is made by averaging the predictions across all trees (described below in 2.4)."

In the methods section, parameters such as the number of nodes per tree (p. 11, line 10), number of trees (p. 10, line 11) and number of variables per split (p. 11, line 11) are stated, but the methodology for choosing these numbers is not explained in sufficient detail (or at all, in the case of the number of nodes). The parameter used to select the best settings is described as the "values that produce the lowest test error" – is this error just rate of incorrect identification?

While not explicitly stated previously, the method of hyperparamter optimization described is grid search, or parameter sweep, whereby numerous combinations of the parameters are exhaustively enumerated. The error rate mentioned is the test error, or out-of-bag error (not training error). Each of your points has been clarified in more detail on 12:1-5:

"Using grid search, the optimal model was determined by enumerating combinations of these parameters on a coarse grid and selecting the values that produce the lowest test error, or out-of-bag error. Given several lists of parameters, where each list corresponds to a different model hyperparameter, models are trained one-by-one until each combination of parameters has been tested. For this study, the grid representing variables per split was spaced by 1 and the grid for number of trees was spaced by 5. The number of nodes in each tree depends on other hyperparameters and cannot be explicitly set."

On p. 11, line 18, the noun asymptote is used as a verb. The sentence should be rewritten.

**We have substituted "converges" for "asymptotes" on 12:10.**

On pp. 20 - 21, the authors illustrate the advantages of their method by mentioning that an unexpected contaminant was detected based on the results. The implication is that this is possible using their method but not others, however a distance metric-based algorithm would likely also be able to identify this contaminant, as it contained additional peaks. The authors should clarify how this example specifically illustrates the strength of their method (if it does).

The reviewer is correct that the contaminant would have been identified with other techniques but here was identified as a direct result of feature ranking. The implication has been clarified on 21: 10-17

"In this particular study, the contaminant was identified and removed in the dimensionality reduction step while reasoning through the subset of ranked features. As illustrated by Figure 2, cobalt is suspiciously identified as the second most important variable for classification, but it is a known component of dry powder dispersion equipment used on some samples. The contaminate peak would be present in a cluster analysis, but it would not be obvious to pick out and remove as standard clustering is not typically suited for variable rankings. "

Figure 4 illustrates a comparison of results using the random forest and a distance classifier. However, no information is provided about the (user-defined) parameters used to define different clusters in the distance metric example, making this comparison tricky. If the parameters were changed slightly, these results would likely vary...

We do not believe the results vary using this method; In the context of aerosol classification, clustering is first used to find the cluster centers of an unlabeled dataset. "Classification" is then done by manually labeling each cluster before assigning unknown aerosols to the nearest cluster center using some distance metric. While user-defined parameters strongly influence the behavior of the clustering algorithm, the assignment of unknown aerosols after convergence does not depend on parameters. The distance-based classifier is uniquely defined by the distance metric (Euclidean in this case, although cosine similarity is also used in the literature) and the input data. In our case, the centers are the mean of each aerosol type, representing a simple baseline classifier to compare results against. To draw an analogy with clustering, the assumption in that some clustering algorithm has already converged to the center of each aerosol category.

...Also, the labels of a) and b) should be removed from the figure caption; top and bottom row are sufficient. The figure would be more useful if the algorithm type were included in the labels for the specific matrices, so that one needn't rely on the text in the figure caption to identify what the matrices represent. Maybe replace "Aerosol Confusion Matrix (Positive)" with "Random Forest (Positive)" or "Euclidian Distance (Positive)" for clarity.

**Figure 4 has been modified as suggested.**

Figure 5 is still confusing, in that it shows the ~1/3 of particles (soot) that are introduced into the AIDA chamber but which the PALMS instrument cannot detect. The figure caption suggests that the instrument transmission efficiency is discussed in the text, but that discussion (p. 18, lines 18 - 21) is very brief and is mostly directed towards explaining the significant undercounting of the larger particles. This discussion should be expanded, and ideally, the data presented in the figure should be shown corrected for the inlet transmission. As it stands now, the use of the pie charts only illustrates that the match between the concentration (it is not specified whether the input aerosol in the chamber is given in number concentration or mass concentration, although presumably the PALMS results are provided in number concentration) is poor. Figure 5 has been revised based on both reviewers' comments and now shows a full particle pie chart and one within the PALMS instrument detection range. All results are given as a relative concentration in terms of number, and not mass, the relevant quantity for single particle instruments (such as PALMS), and this is now explicitly stated in the caption. The impact of inlet transmission is referenced in the figure caption to Cziczo et al., 2006 which discusses this and provides limits.

The specificity with which the different particle types can be identified is sufficiently different in positive and negative ion spectra to warrant more discussion than is given. Overall, the data presented in this figure cannot serve to make the readers of this paper confident that the picture of the aerosol composition obtained by these experiments would do an excellent job of representing the reality of what is present.

The reviewer is correct and the blind results section (pg. 19-21) has been extended to discuss uncertainties and potential biases in more detail.

**Reviewer 2**

The authors present a new analytical tool to tackle the difficult task of analyzing datasets generated by single-particle-laser-ablation-mass-spectrometry (spms). They utilize the random forest as a machine learning approach. The authors state why they apply this method and what they expect. It is clearly presented how a random forest is generated and subsequently used.

The produced results are scientifically promising. And grant a novel view onto these kind of datasets.

After building the random forest and analyzing its properties, the authors apply the forest to a blind dataset. The results are shown and they differ quite significantly from the assumed true constitution. (Fig 5).

The critical discussion of these results and of the general problems of supervised machine learning remains quite limited. The most important neglected point being dataset bias. E.g. the random forest might find hidden correlations within the training dataset that have nothing to do with the chemistry of the particles but with instrumental parameters, and which most probably are not apparent during the blind test.

One example would be that the signal intensities could depend on ambient temperature, ambient pressure, or laser power. Especially result showing close to 100% true classifications, should be a examined more critically than done by the authors.]

We repeat the overview statements for clarity. We agree with the dataset bias point and note that it is repeated below; In response to that point the issue of dataset bias is discussed in full detail at the end of the blind dataset section 20: 16 - 21: 2.

Following a list of individual remarks:

p.5\_16 chemically similar and easily separable this is an oxymoron chemically similar implies a strong overlap of chemical features

We meant to convey that broad categories are easier to separate in feature space. The correction has been made on 5:21.

p.8\_20-23 why is this normalization done this removes information about the ionization efficiencies, how can you differentiate between ionization efficiency and relative abundance

The reviewer is correct that ionization efficiency is one factor in generation of ions but the topic of aerosol ablation and ionization in single particle mass spectrometers is more complex. Other factors include, but are not limited to, mixing state, matrix effects, hydration state, aerosol position in the desorption and ionization laser, etc. Each factor, as well as their interplay, is the topic of multiple papers. Normalization is commonly used to compare spectra to each other since these factors may vary for each particle (i.e., from spectra to spectra).

We discussed this comment and have decided the succinct response is to clarify this with "Mass peaks represent fractional ion abundance, measured as a total signal (ion current) normalized to allow for spectra to spectra comparison [Cziczo et al., 2006]."

p.9\_18 Are there really up to 3000 tests before reaching a node? This would mean each m/z value is tested roughly 6 times. And the tree would have to be at least 6000 nodes.

The maximum depth of a tree (number of nodes before reaching a leaf) ultimately emerges from other model parameters that were optimized. It is noted that most trees, and most paths to a given leaf, will have significantly fewer than 3000 tests (10 - 3000). Because there are strongly overlapping aerosol types such as the fertile soils, it is not unreasonable for a tree to occasionally require ~3000 test . For example, if the aerosol types are not perfectly separable along a given dimension, the algorithm will continue to create nodes (which encoding a line that separates the two types) that slowly converge to a solution that best separates either category.

p.10\_19 Should be left out in ~40% of the trees. Here would be a good point to mention dataset bias. Because although a spectrum is not in the training-set there could still be hidden correlation to the others.

The issue of dataset bias, including this point, is discussed in detail at the end of the blind dataset section 20: 16 - 21: 2.

p.11\_4 might be better to follow if "To generate variability in the model only a random set of splits is tested at each node and only the best split in terms of entropy is chosen"

We agree this reads more smoothly. The sentence on 11: 13 -15 has been updated.

p.12\_15 markers

This has been corrected.

p.13\_11 helps to put

This has been corrected.

p.16\_18-19 I don't understand Point b). If it is distinct why is it not separated.

We are conveying that aerosols with similar properties can appear mathematically distinct when clustering with a distance metric. Aerosols within a broader category can still occupy distinct regions of parameter space, a consequence that leads to the need to manually combine clusters in previous studies as mentioned on 5: 16-18. In Figure 1, for example, bacteria (yellow) forms two distinct clusters. With a distance metric, spectra in the smaller bacteria cluster will likely be clustered with the collocated hazelnut particles rather than the primary bacteria cluster center in the vicinity of (.01, .001). The issue is compounded when larger, more chemically diverse categories are defined.

p.16\_20 Here is an example of dataset bias and it is shown to hold some information but the backside is not discussed.

Please note that this is also commented on in the overview remarks. We agree and expand the issue of dataset bias which is discussed in detail at the end of the blind dataset section 20: 15 - 21: 2.

p.18\_2-9 If Misclassification is as shown 1-4% it cannot explain the (not SOA) fractions of 3-9% for fertile soil, ATD and cellulose. There must be an additional source of error.

Experimental uncertainties such as internal mixing and transmission efficiency, as well as model uncertainties (including overfitting) explain additional differences between the test error and generalization error. Details regarding these uncertainties have been extended at various points on pages 19-21 (please see the full track changes version), and a discussion of dataset bias has been added on 20: 15 - 21: 2.

p.18\_13 The authors state that 90% of the mixture can be characterized with most certainty. Comparing this to Fig 5. this statement seems quite exaggerated.

Since the criterion for "most certain" was relative and loosely defined, we have updated the statement and removed the mention of (~90%) on 19: 5 – 7.

"Since there is significant model agreement on the percentages of SOA and coated feldspars, this part of the blind mixture population can be characterized with more certainty."

p.18\_16 It seems unrealistic that there was so much effort put into this campaign and without characterizing the used aerosols in more detail.

We have expanded the paragraph to provide our full report from the AIDA facility in the revised paragraph (extending to the comment below) : "The aerosols reported in the blind mixture were soot, mineral dust, and SOA. The soot aerosols used in the blind study were smaller than in the training data experiments and were below the cutoff diameter for PALMS; they were therefore not detected and therefore could not be identified by the algorithms. This bias is transmission efficiency should be noted, whereby aerosols are detected at a rate that depends on their size and aerodynamic properties [Cziczo et al., 2006]. The result is that particles with diameters below ~200 nm or greater than ~1000 nm are detected with increasing inefficiency which lead to relative undercounting of small soot or large mineral dust [Cziczo et al., 2006]. The specific mineral component was not identified and may have been either a pure mineral or soil dust. Both algorithms robustly labeled SOA with large agreement, consistent with the 100% accuracy observed in the test set. "

p.18\_18 Why was the PALMS instrument able to see soot particles in the training set with 100% accuracy if it cannot be seen? Why not use the size distributions of the individual components

and the transmission efficiency of the PALMS to at least get the expectable aerosol constitution?

The soot in the blind experiments was smaller than in the training data set and we regret this was not explicitly stated earlier. We clarify now as "
[revised manuscript text omitted]
                                                                                       |
| 35           | 35CI                                                                                                   | 23          | Na +                                                                                  | 35                        | 35 Cl -                                                                                                 | 23          | Na ⁺                                                                               |
| 25           | C 2 H -                                                                          | 59          | Co +(1) /CaF + /                                                           | 26                        | $CN^{T}/C_{2}H_{2}^{T}$                                                                                                       | 59          | $Co^{+(1)}/CaF^{+}/C_{2}H_{2}OOH^{+}$                                                         |
|              |                                                                                                        |             | $C_2H_2OOH^+$                                                                                    |                           |                                                                                                                               |             |                                                                                               |
| 24           | C 2                                                                                         | 39          | 39 K +                                                                     | 46                        | NO 2                                                                                                               | 44          | SiO + /COO + / 44 Ca + /AIOH +         |
| 57           | C 2 OOH -                                                                        | 12          | C +                                                                                   | 1                         | H                                                                                                                             | 39          | 39 K +                                                                  |
| 59           | C 2 H 2 OOH -
/AlO 2 -                       | 24          | C 2 +                                                                      | 57                        | C₂OOH -                                                                                                            | 28          | Si + /CO +                                                              |
| 43           | HCN - /AIO -                                                                     | 41          | 41 K + /C 3 H 5 +                         | 59                        | C 2 H 2 OOH * /AlO 2 *                                                 | 41          | 41 K + /C 3 H 5 +                      |
| 1            | H                                                                                                      | 204-
208 | Pb region ( 204 Pb,
206 Pb, 207 Pb and
208 Pb) | 45                        | COOH                                                                                                                          | 54          | 54 Fe +                                                                 |
| 26           | $CN^{T}/C_{2}H_{2}^{T}$                                                                                | 27          | $AI^{+}/C_{2}H_{3}^{+}$                                                                          | 42                        | CNO - /C 2 H 2 O -                                                                | 56          | $Fe^{+}/CaO^{+}$                                                                              |
| 46           | NO 2                                                                                        | 44          | SiO + /COO + / 44 Ca + /Al
OH +        | 43                        | HCN - /AIO -                                                                                            | 27          | $AI^+/C_2H_3^+$                                                                               |
| 16           | 0-                                                                                                     | 57          | $^{57}$ Fe + /CaOH + /C 3 H 4 O
H +    | 16                        | 0                                                                                                                             | 45          | SiOH + /COOH +                                                          |
| 17           | OH                                                                                                     | N/A         | aerodynamic                                                                                      | 73                        | C 2 O 3 H - /                                                                                | 66          | Zn⁺                                                                                           |
|              |                                                                                                        |             | diameter                                                                                         |                           | C 3 H 2 OOH 3 -                                                                   |             |                                                                                               |
| 61           | SiO 2 H /29 SiO 2
/C 5 H - /CHO 3 | 83          | $H_3SO_3^+/C_4H_2OOH^+$                                                                          | 63                        | PO 2                                                                                                               | 57          | 57 Fe + /CaOH + /C 3 H₄OH +            |
| 63           | PO 2                                                                                        | 87          | 87 Rb + /CaPO +                                                 | 60                        | SiO 2 - /C 5 - /CO 3 - /
AlO 2 H - | 87          | 87 Rb + /CaPO +                                              |
| 19           | F - /H 3 O -                                                          | 13          | CH⁺                                                                                              | 15                        | NH - /CH 3 -                                                                                 | 85          | 85 Rb +                                                                 |
| 76           | SiO 3                                                                                       | 66          | Zn⁺                                                                                              | 24                        | C2                                                                                                                            | 83          | $H_3SO_3^+/C_4H_2OOH^+$                                                                       |
| 77           | SiO 3 H - / 29 SiO 3 -                          | 28          | Si + /CO +                                                                 | 76                        | SiO 3                                                                                                              | 24          | C 2 +                                                                   |
| 79           | PO 3                                                                                        | 85          | 85 Rb +                                                                    | 32                        | 0 2                                                                                                                | 204-
208 | Pb region ( 204 Pb, 206 Pb,
207 Pb and 208 Pb) |
| 60           | SiO 2 /C 5 /CO 3
/ AIO 2 H                              | 72          | $FeO^{+}/CaO_{2}^{+}$                                                                            | N/A                       | aerodynamic
diameter                                                                                                       | 40          | Ca +                                                                               |
| 45           | COOH                                                                                                   | 54          | 54 Fe +                                                                    | 71                        | C 3 H 2 OOH -                                                                                | 153         | 137 BaO +                                                               |
| N/A          | aerodynamic
diameter                                                                                | 82          | ZnO +                                                                                 | 50                        | C 4 H 2                                                                                                 | N/A         | aerodynamic diameter                                                                          |

1 Contamination

2 Table 2. Features rankings for differentiation of particles between labels and between

3 broad categories in positive and negative ion modes. See text for additional details.

4

| Category         | Negative | Postive | Category         | Negative            | Posti         |
|------------------|----------|---------|------------------|---------------------|---------------|
| Fertile Soil     | 0.88     | 0.83    | Fertile Soil     | $0.024\ {\pm}0.020$ | $0.035 \pm 0$ |
| Mineral/Metallic | 0.93     | 0.98    | Mineral/Metallic | $0.017 \pm 0.027$   | $0.006 \pm 0$ |
| Biological       | 1.00     | 1.00    | Biological       | 0.000               | $0.001 \pm 0$ |
| Other            | 0.96     | 0.93    | Other            | $0.021\ {\pm}0.015$ | $0.024 \pm 0$ |

2 Table 3. Model performance by category and ion mode on a population consisting entirely of aerosols within that category. Left: Average classification accuracy where 1.0 3 = 100% precision (Powers, 2007). Right: mean and standard deviations of 4 misclassification. 5

6